# ATTENTIVE EXPLAINABILITY FOR PATIENT TEMPORAL EMBEDDING

## ABSTRACT

Learning explainable patient temporal embeddings from observational data has mostly ignored the use of RNN architectures that excel in capturing temporal data dependencies but at the expense of explainability. This paper addresses this problem by introducing and applying an information theoretic approach to estimate and quantify the degree of explainability of such architectures. Using a communication paradigm, we formalize metrics of explainability by estimating the amount of information that an AI model needs to convey to a human end user to explain and rationalize its outputs. A key aspect of this work is to model human preferences and prior knowledge at the receiving end and measure a lack of explainability as a deviation from these human preferences. We apply this paradigm to medical patient representation problems by regularizing loss functions of temporal autoencoders according to the derived explainability metrics to guide the learning process towards models producing explainable outputs. We illustrate the approach with convincing experimental results for the generation of explainable temporal embeddings for critical care patient data.

## 1 INTRODUCTION

The success of AI models to solve real world problems has been hindered by the inability to fully understand the complex mechanisms behind inferences produced by these models. This problem is exacerbated in healthcare where clinicians and other stakeholders consuming AI model outputs need to understand the rationales behind the underlying AI inference processes. Learning explainable representations for patient data has proved to be notoriously difficult and has slowed the adoption of powerful sequence to sequence architectures for this task. In Choi et al. (2016a), the authors describe an interesting and effective approach for the embedding of medical patient data. However, they avoid RNN like architectures probably able to produce more accurate representations but at the expense of explainability.

Explainability in AI in general has been receiving a great deal of attention within the academic community with numerous methodological advances for the learning of models that are easier to digest by humans. In Ribeiro et al. (2016) and Ribeiro et al. (2018), the authors proposed to locally learn simple linear or tree models to approximate the local performance of complex classifiers. Their approach assumes that explainability is heavily related to model complexity. In Choi et al. (2016b), the authors proposed a Recurrent Neural Network architecture able to compute attention coefficients linking the predictions of the model back to the raw features used by the model at different time points. In this case, explainability is also a local concept that relates to attention mechanisms justifying the predictions made by a model by pointing back to the source data deemed responsible for the predictions. Recently in Dhurandhar et al. (2018), the authors proposed to relate explainability not only to the presence of input features responsible for an inference but also to the omission of input features (e.g., the man who does not wear glasses). In Koh & Liang (2017), the authors proposed to use influence functions to understand the internals of an AI model by tracing back predictions to training data instances that have influenced it.

While these papers cleverly shed light on how complex AI models are behaving, they all use different informal notions of explainability–may it be complexity, attentiveness linking predictions to input features or training data influence on predictions. They all treat explainability as a qualitative characteristic of AI models and do not rely on well defined measurable aspects of explainability. In

this paper, we treat aspects of explainability as quantitative measurable properties of models and propose approaches to measure the degrees of explainability of learned models in a regression setting, with application to computational embedding. In general, formalizing explainability is quite hard. In this work, we only focus on specific aspects of it that are grounded with the following principles:

- *Observability:* We restrict our efforts to the generation of explanations as seen by an external observer witnessing the input-output relationship for a given AI system. This principle is restrictive as it treats any AI agent as a black box. However, we deem it to be a reasonable assumption that humans commonly make, specially in physical sciences when modeling complex physical phenomena from observational data.

- *Communicability:* We consider explainability to be a communication problem, with a dialog between an external AI observer and a rational human actor with the external AI observer explaining the behavior of the AI system to the rational human actor at the receiving end of this communication. The amount of information that needs to be produced by the AI external observer for explainability is consequently inversely proportional to the degree of explainability of the AI system.

- *Subjectivity:* We view explainability as a subjective concept that depends on the mind set of the receiving rational human actor, in terms of his/her preferences and amount of prior knowledge that he/she may have acquired. This principle prompts us to model some notion of preferences and/or assumed prior knowledge available to these human actors with prior distributions. In these cases, AI models are deemed less explainable as they produce predictions based on facts that are harder to explain using generally accepted human prior knowledge.

Based on these principles, we model explainability using information theoretic constructs measuring the amount of information that need to be transmitted by an AI agent to a human end user to explain the rationales behind the predictions. The main contributions of this paper are three fold: (i) an information theoretic framework to effectively measure aspects of explainability, (ii) the application of the framework to regression problems using attention mechanisms, and (iii) conclusive experimental results on patient temporal embedding problems. The rest of the paper is organized according to these three contribution areas.

## 2 EXPLAINABILITY WITH THE MINIMUM DESCRIPTION LENGTH PRINCIPLE

### 2.1 NOTATIONS

Throughout the paper, scalars are represented with lower case letters in a non bold typeface, e.g., $x$. Sets are represented with upper case letters in a calligraphic typeface, e.g., $\mathcal{H}$. Vectors are denoted in bold lower case letters e.g., $\mathbf{x} \in \mathbb{R}^k$ from some positive $k \in \mathbb{N}$. Matrices are represented in bold upper case letters, e.g., $\mathbf{A} \in \mathbb{R}^{n \times m}$, where both $n, m \in \mathbb{N}$. In general, we use subscripts to index elements in a vector or in a matrix. For instance, $\mathbf{x}_i$ is a scalar corresponding to the $i$th element of vector $\mathbf{x}$ while $\mathbf{A}_{i,j}$ denotes the $i, j$ entry of matrix $\mathbf{A}$. The data elements in this paper are always attached to a specific entity (e.g., a patient in healthcare). To avoid notation clutter, we often drop these references to entities and describe the methodology at the entity level. The data for a given entity is represented by a matrix $\mathbf{X} \in \mathbb{R}^{n \times m}$. A row $\mathbf{X}_i, 1 \leq i \leq n$ represents an $m$ dimensional feature vector collected at discrete time $i$. Similarly, the output of an AI model processing $\mathbf{X}$ is represented by the matrix $\mathbf{Y} \in \mathbb{R}^{n \times k}$ corresponding to a discrete time series of $k$ dimensional output vectors.

### 2.2 THE MINIMUM DESCRIPTION LENGTH PRINCIPLE

The Minimum Description Length principle is a formalization of the Occam Razor principle guiding the search for good models that explain data using description lengths (Grünwald (2007),Rissanen (1998)). A simple version of it uses a two-part coding approach to select the "best" model $M^{opt}$ as the one that minimizes the sum of the length in bits for a description of the model (part one of the code) and the length in bit of the effective description of the data using using the model.

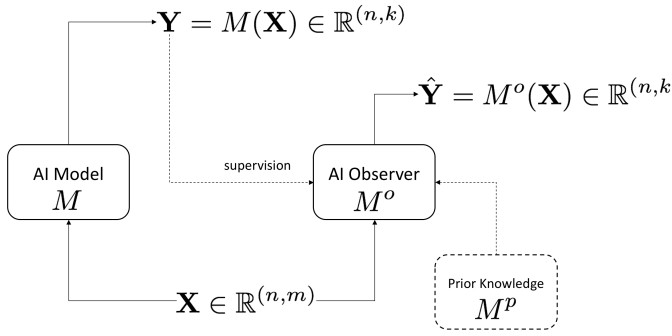

Figure 1: Conceptual explainability model with an external observer model $M^o$ explaining the behavior of an AI model $M$ in the presence of prior $M^p$.

Let $\mathcal{M}$ denote the set of all computable models in the Turing sense. The MDL defines the best model as:

$$M^{opt} := \min_{M \in \mathcal{M}} \left( L(M) + L(\mathbf{X}|M) \right) \tag{1}$$

where $L$ is a function that returns description lengths. The MDL approach imposes preferences for models that balance their complexity with their goodness of fit. The first part of the code is essentially the *model complexity* while the second part measures the *goodness of fit* for the model on the data. An idealized version of the MDL has been defined in Vitányi & Li (2000) where $L$ is replaced with $K$, the Kolmogorov complexity. However, this version is not computable in the Turing sense. In the following sections, we present practical methods to compute estimate $L$ for explainability purposes.

## 2.3 EXPLAINABILITY METRICS

Following the observability and communicability assumptions described in Section 1 and the MDL principle, we measure the explainability of an AI model $M$ with description lengths of an external observer model only able to perceive the inputs and outputs of $M$ to model this relationship (See Figure 1). In this setting, an explanation $M^o$ for an AI model $M$ is essentially such an observer model describing how its inputs $\mathbf{X}$ and outputs $\mathbf{Y}$ are related in a language that is understandable by human actors. Assuming the presence of no additional knowledge for $M^o$ to characterize this input output relationship, we measure the effectiveness of $M^o$ with two part codes as follows:

**Definition 1.** The explainability provided by an observer $M^o$ of model $M$ is defined as:

$$e^{ag}(M^o, M, \mathbf{X}) := L(M^o) + L(M(\mathbf{X})|M^o, \mathbf{X}) \tag{2}$$

The knowledge agnostic explainability of a model $M$ is defined as:

$$E^{ag}(M, \mathbf{X}) := \min_{M^o \in \mathcal{M}} e^{ag}(M^o, M, \mathbf{X}) = \min_{M^o \in \mathcal{M}} [L(M^o) + L(M(\mathbf{X})|M^o, \mathbf{X})] \tag{3}$$

As stated in Section 1, we also assume that explainability can be subjective and dependent on the mind set of receiving rational human actors, in terms of the amount of prior knowledge that they have acquired. In our framework, this prior knowledge is represented as a prior model $M^p$ that these human actors would use to explain the $(\mathbf{X}, \mathbf{Y})$ relationship. For instance, an expert physician may explain the general loss of health of a critical care patient from features representing reductions in heart rate variability in the source data. For this physician, an AI model producing rationales that are inline with this prior knowledge would certainly by easier to explain than other AI models using variables that are further from her/his mental mind set. Consequently, we modify the complexity term of the previous definition to come up with the following measures of explainability in the presence of prior knowledge.

**Definition 2.** With a computable prior knowledge $M^p$, the explainability provided by an observer $M^o$ of model $M$ is defined as:

$$e^{aw}(M^o, M, M^p, \mathbf{X}) := L(M^o|M^p) + L(M(\mathbf{X})|M^o, \mathbf{X}) \tag{4}$$

The knowledge aware explainability of a model $M$ is defined as:

$$E^{aw}(M, M^p, \mathbf{X}) := \min_{M^o \in \mathcal{M}} e^{aw}(M^o, M^p, M, \mathbf{X}) = \min_{M^o \in \mathcal{M}} [L(M^o|M^p) + L(M(\mathbf{X})|M^o, \mathbf{X})] \tag{5}$$

While we have not put any restrictions on the class of models for $M^o$, practical computability considerations prompt us to focus on models that can be algorithmically derived and with structural properties that are understandable by humans (e.g., regression trees, linear regression models).

## 2.4 Explainability with Attention Models for Regression Problems

$E^{ag}$ and $E^{aw}$ are not computable in the general Turing sense of the word. However, with restrictions on the class $\mathcal{M}$ of models, approximations for these quantities can be defined. In practice, these restrictions are typically dictated by the problem at hand. In the rest of this paper, we focus exclusively on regression problems, thus allowing us to tackle the medical patient embedding problem that we are interested in. Consequently, we restrict $\mathcal{M}$ to the set $\mathcal{M}_{reg}$ of regression models for all models $M$ that estimate $\mathbf{Y}$ from $\mathbf{X}$, and all observers $M^o$ that estimate the $\hat{\mathbf{Y}} = M(\mathbf{X})$ from $\mathbf{X}$.

With the popularity and effectiveness of attention mechanisms to provide input/output rationales, we further impose attention mechanisms in the architecture of our observers. This additional step not only allows us to measure the complexity of our observers but also allows us to produce effective explanations. More specifically, our observers make use of local attention models estimating at each time $i$ the attention that $M$ is applying to each input feature $\mathbf{X}_{ij}$. We define $\alpha_i(\mathbf{X}_i) = \text{SoftMax}(\Psi(\mathbf{X}_i)) \in \mathbb{R}^m$, where $\Psi$ is an arbitrarily complex model producing attention coefficients. Dense layers are typically used to compute $\Psi$ but more complex recurrent neural architectures to compute attention coefficients can also be used as suggested in Choi et al. (2016b). $M^o$ leverages the $\alpha_i$'s in the following way:

$$M^o(\mathbf{X}_i) = \mathbf{W}\alpha_i \odot \mathbf{X}_i = (\mathbf{W}[:, k] \odot \alpha_i)\mathbf{X}_i^T = \mathbf{P}\mathbf{X}_i^T \tag{6}$$

where $\mathbf{P} = \mathbf{W}[:, k] \odot \alpha_i$ with $\odot$ denoting the Hadamard product and $\mathbf{P} = \mathbf{W}[:, k] \odot \alpha_i$ being a matrix obtained by stacking the results of the Hadamard product of columns of $\mathbf{W}$ with $\alpha_i$.

### 2.4.1 Estimating The Goodness of Fit for Regressive Observers

With this restriction to $\mathcal{M}_{reg}$, we model the regression error with a zero mean normal distribution. This is quite common in regression analysis and also in the application of the MDL to regression problems (Grünwald (2007)). Hence, for all $M^o \in \mathcal{M}_{reg}$, $\mathbf{Y} = M^o(\mathbf{X}) + \boldsymbol{\epsilon}$, where $\boldsymbol{\epsilon}_j \sim N(0, \sigma^2)$ for $1 \leq j \leq k$ This restriction allows us to approximate the goodness of fit term $L(M(\mathbf{X})|M^o, \mathbf{X})$ in Equations 2 and 4 for any regressive observer for any AI model without any further assumption on $M$. Since this term essentially measures the amount of information that cannot be explained by the model $M^o$, it tends to have random properties and can be approximated using its Shannon-Fano code (Cover & Thomas (2006), Grünwald (2007)):

$$L(M(\mathbf{X})|M^o, \mathbf{X}) \approx -\log p(M(\mathbf{X})|M^o(\mathbf{X}), \mathbf{X}) = -\sum_{i=1}^{k} \log p(M(\mathbf{X})_i|M^o(\mathbf{X}), \mathbf{X}) \tag{7}$$

using the conditional independence of each $(M(\mathbf{X}))_i$ given $\mathbf{X}$. Clearly, for each $i$, $p(M(\mathbf{X})_i|M^o(\mathbf{X}), \mathbf{X})$ follows also a Gaussian distribution $N(M^o(X)_i, \sigma^2)$. Hence, the goodness of fit can be rewritten as a scaled mean squared error between $M(\mathbf{X})$ and $M^o(\mathbf{X})$:

$$L(M(\mathbf{X})|M^o, \mathbf{X}) = \frac{nk}{2}\log(2\pi\sigma^2) + \frac{1}{2\sigma^2}\sum_{j=1}^{k}\sum_{i=1}^{n}(M(\mathbf{X})_{ij} - M^o(\mathbf{X})_{ij})^2 \tag{8}$$

### 2.4.2 Estimating the Complexity of Linear Regressive Observers

As stated in Grünwald (2007), the basic MDL principle does not define the expression of the complexity terms in Equations 2 and 4. These terms are very much dependent on the restricted class of observer models and not on any properties of the data $\mathbf{X}$. From Equation 6, we are restricted to $M^o(\mathbf{X}) = \mathbf{P}\mathbf{X}^T$, a multiplication of the input with the $m$ by $k$ matrix $\mathbf{P}$. Rows of $\mathbf{P}$ can be scaled into attentive distribution that we use to estimate the complexity of $M^o$ by measuring the compactness of these rows. Intuitively, the more compact these row vectors are, the easier it is to explain the corresponding predictions produced by $M$ since the observer would hypothetically need to communicate less bits of information to describe how inputs $\mathbf{X}$ relate to outputs in $M$. A natural

measure of compactness is the Shannon entropy and we estimate the model complexity as follows: Let $\mathbf{Q}$ be a matrix such that $\mathbf{Q}_j = \text{SoftMax}(|\mathbf{P}_j|)$, then:

$$L(M^o) \approx \frac{1}{k} \sum_{j=1}^{k} H(\mathbf{Q}_j) \tag{9}$$

where $H(p) = \sum_i p_i \log \frac{1}{p_i}$

With a conditional on preferences or prior knowledge $M^p$, one may be tempted to measure $L(M^o|M^p)$ with a conditional entropy, thus extending naturally the approach that we used to estimate $L(M^o)$. However, the computation of such conditional entropy is ill defined as it requires an unknown joint distribution. Instead, we complement $L(M^o)$ with the KL divergence between the attentions $\mathbf{Q}_j, 0 \leq j \leq k$ and their counterpart computed from prior preferences or knowledge $\boldsymbol{\beta}_j, 0 \leq j \leq k$.

Consequently,

$$L(M^o|M^p) \approx \frac{1}{k} \sum_{j=1}^{k} (H(\mathbf{Q}_j) + D_{KL}(\mathbf{Q}_j||\boldsymbol{\beta}_j)) = \frac{1}{k} \sum_{j=1}^{k} \mathbf{Q}_j . \log \boldsymbol{\beta}_j \tag{10}$$

In this case, the complexity becomes a simple average cross entropy across all $k$ dimensions of the output.

## 3 Learning Explainable Temporal Embeddings

Despite the large-scale adoption of Electronic Health Records (EHR) by medical institutions, the secondary re-use of these data sets and its impact in healthcare has been lukewarm at best. Beyond all the data governance challenges that need to be overcome for proper access and integration, these data sets are quite raw, with very complex data models. They always need to be refined to become ready for analysis and these data refinement steps are quite expensive, often tailored for specific applications and not reusable. Deep representation learning techniques have been proposed by several research groups to tackle these challenges. However, most of these approaches refrain from using RNN architectures in attempts to preserve the interpretability of the learned models at the expense of being able to encode efficiently the temporal patterns in the data. Using external regressive operators as observers, we address this issue and present in this section an RNN based framework for the representation learning of temporal embeddings.

### 3.1 An RNN Pipeline for Temporal Embeddings

Figure 2 illustrates the proposed deep learning architecture. At a high level, this architecture is an auto-encoder consisting of three components. First, an RNN encoder $M^{enc}$ is used to transform the input data $\mathbf{X}$ into embeddings $\mathbf{Y}$. We experimented with various RNN cells for the encoder including the popular LSTM and GRU cells. We eventually settled on the minimalRNN cell presented in Chen (2017) and Chen et al. (2018) because of its predictable dynamics as it is designed to rule out mixing effects across dimensions of the embeddings produced by our autoencoder. This choice allows us to improve the likelihood of developing a regressive observer model with a reasonable goodness of fit without sacrificing too much on the quality of the embeddings.

The second component is an RNN based decoder $M^{dec}$ that attempts to reconstruct the original data $\mathbf{X}$ from the output of the encoder. We have equipped this decoder with a temporal attention scheme similar to the ones typically used in NLP sequence 2 sequence architectures for machine translation. This temporal attention allows the decoder to reconstruct $\mathbf{X}_i$ using embeddings already computed at previous times $i - 1$ to $i - w$, $w$ being a hyper-parameter that we keep small (i.e., less than 10).

The third component of this deep learning architecture is the regressive observer $M^o$ described in the previous section and used to track the inputs and outputs of the encoder to provide and estimate explainability. This component is taking $\mathbf{X}$ as input and producing an estimate $\hat{\mathbf{Y}}$ of the embedding $\mathbf{Y}$. During training, this component may have access to a prior model of user preferences $M^p$ that it may use in regularization steps to control deviations from these preferences. $M^p$ allows us to control the contribution of input variables into each dimension of the embeddings produced by the encoder.

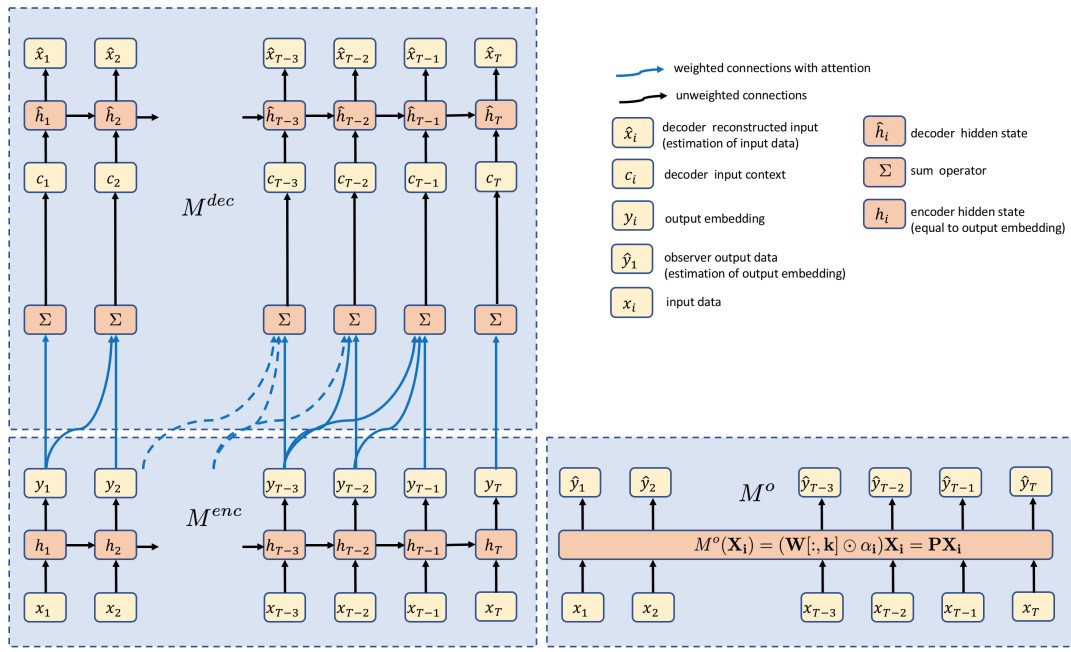

Figure 2: Attentive RNN based embedding

To train this deep learning architecture, we use a multi-task approach where we train each of these components jointly. The joint training of this autoencoder with the observer allows us to softly enforce explainability constraints in this pipeline. While the role of the observer is essentially to estimate the degree of explainability, we use it here to constrain the generation embeddings towards explainable models according to the metrics that we defined in the previous section. The encoder and decoder are trained together to minimize the mean square reconstruction error between $\mathbf{X}$ and $\hat{\mathbf{X}}$, according a loss $\mathcal{L}^{rec}$ expressed as:

$$\mathcal{L}^{rec} = \frac{1}{nk} \sum_{i=1}^{n} \sum_{j=1}^{k} ((\mathbf{X})_{ij} - (\hat{\mathbf{X}})_{ij})^2 \qquad (11)$$

The regressive observer is trained to provide the best explanations in the MDL sense. In the absence of prior preferences, its learning is associated with a loss $\mathcal{L}^{ag}$ defined as:

$$\mathcal{L}^{ag} = e^{ag}(M^o, M^{enc}, \mathbf{X}) \approx \frac{1}{k} \sum_{j=1}^{k} H(\mathbf{Q}_j) + \frac{1}{2\sigma^2} \sum_{j=1}^{k} \sum_{i=1}^{n} (M(\mathbf{X})_{ij} - M^o(\mathbf{X})_{ij})^2 \qquad (12)$$

Please note that we dropped the term $\frac{nk}{2} \log(2\pi\sigma^2)$ from the formal expression of $e^{ag}(M^o, M^{enc})$ since this term does not have impact on the optimization of the loss function when $\sigma$ is estimated. In the presence of prior knowledge, the learning of the $M^o$ is associated with a loss $\mathcal{L}^{aw}$ defined as:

$$\mathcal{L}^{aw} = e^{aw}(M^o, M^{enc}, \mathbf{X}) \approx \frac{1}{k} \sum_{j=1}^{k} \mathbf{Q}_j . \log \boldsymbol{\beta}_j + \frac{1}{2\sigma^2} \sum_{j=1}^{k} \sum_{i=1}^{n} (M(\mathbf{X})_{ij} - M^o(\mathbf{X})_{ij})^2 \qquad (13)$$

Putting all these terms together allows us to define the overall loss function for this architecture:

$$\mathcal{L} = \begin{cases} \mathcal{L}^{rec} + \lambda_{ag}\mathcal{L}^{ag} & \text{in the absence of prior knowledge} \\ \mathcal{L}^{rec} + \lambda_{aw}\mathcal{L}^{aw} & \text{in the presence of prior knowledge} \end{cases}$$

## 3.2 EXPERIMENTAL RESULTS

We have performed experiments to demonstrate the effectiveness of our proposed metrics for patient data embedding using the MIMIC-III Johnson et al. (2016) dataset. For these experiments, we used a

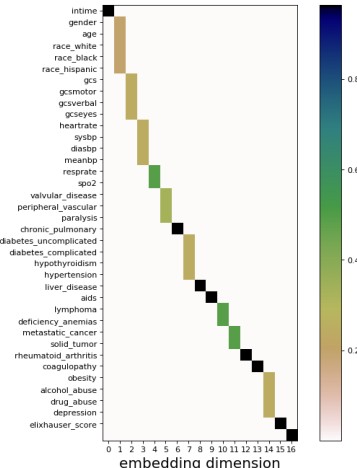

Figure 3: Suggested domain knowledge for the grouping of features into embedding dimensions

cohort of 2395 patients who experienced sepsis, resulting in a total of 109886 data samples. This data set is sampled on an hourly basis. Each data sample consists of 37 variables together with a timestamp and a patient ID. The name of these variables are shown on the Y axis of the plot shown on Figure 3.

The RNN architecture shown in Figure 2 has been implemented on tensorflow v1.5 (Abadi et al. (2015)). Experiments were performed on a cloud environment using a small cluster of K80 GPU units. Various values of learning rate were tried for the training and the results provided in the paper uses a learning rate of 0.0005 with batch size of 10. All results were obtained using a 5 fold cross validation approach with partitions computed at the patient level.

A prior model $M^p$ was gathered manually to impose soft constraints on how input variables may be grouped together to define each of the $k$ embedding dimensions. This prior suggests a uniformly distributed focus for each variable in a given group. This manual grouping led us to define 16 groups that are shown in Figure 3. This figure shows the heat map of a matrix with rows corresponding to input variables and columns corresponding to embedding dimensions. Each column shows a suggested distribution of input variables within an embedding dimension. For instance, the third column suggests a concentration of variables that relates to the Glasgow Coma score (a neurological score estimating the consciousness of patients) while the fourth column suggests a concentration of variables measuring various aspects of the cardiovascular system. Different prior preference models have been tested, providing results that are consistent with the evaluation that we are describing here for the specific $M^p$ shown in Figure 3.

We focused on evaluating two aspects of the system, namely the qualitatively analysis of $M^o$ and the analysis of the impact of explainability regularization on $\mathcal{L}^{rec}$

Figures 4 and 5 qualitatively illustrate the explanations provided by $M^o$. Figure 4 focuses on the knowledge agnostic case and shows heat maps of average attention coefficients for each embedding dimensions for $\lambda_{ag} = 1$ and $\lambda_{ag} = 10$. It is clear from that figure that increasing $\lambda_{ag}$ adds sparsity on the heat maps and help provide more explainable rationales. We also looked at the actual input variables that are selected in each of the dimensions. With $\lambda_{ag} = 10$, most embedding dimensions are dominated by a single variable except for the 13th dimension focusing heavily on the blood pressure variables and the first dimension focusing on respiration rate and mean arterial pressure.

Figure 5 focuses on the knowledge aware case and shows heat maps of the average attention coefficients for each embedding dimensions for $\lambda_{aw} = 1$ and $\lambda_{aw} = 5$. A quick look at these plots shows that the system is indeed attempting to learn explanations that are in accordance with the prior preferences imposed by regularization. As $\lambda_{aw}$ increases, $M^o$ converges more and more towards $M^p$.

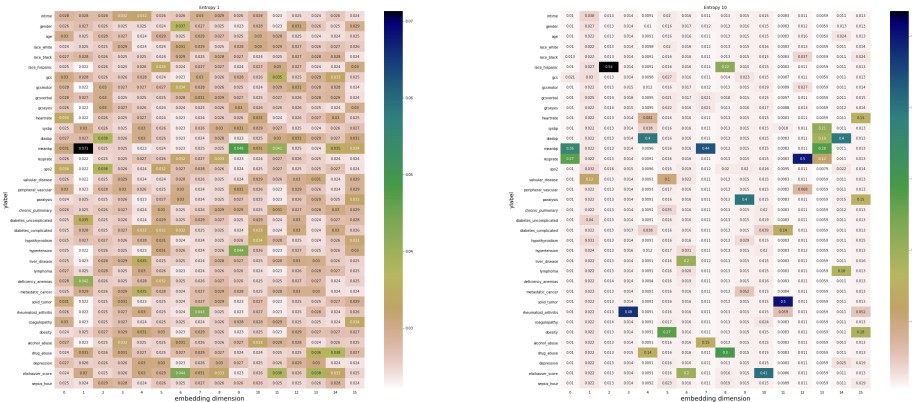

Figure 4: Distribution of the attention coefficients for $\lambda_{ag} = 1$ on the left and $\lambda_{ag} = 10$ on the right.

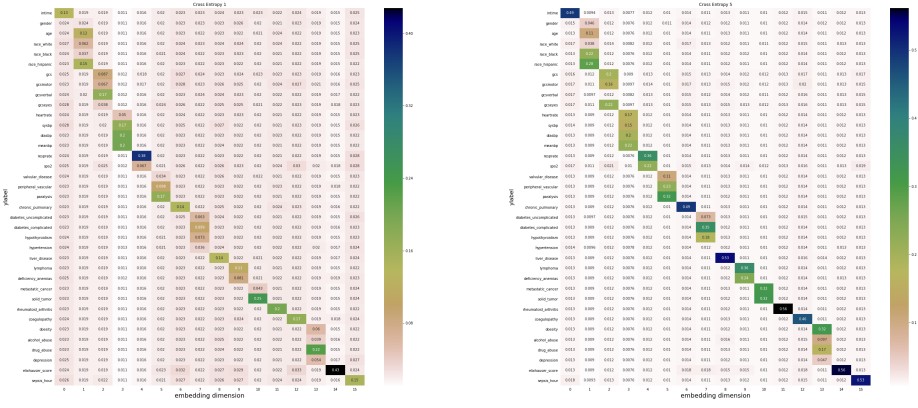

Figure 5: Distribution of the attention coefficients for $\lambda_{aw} = 1$ on the left and $\lambda_{aw} = 5$ on the right.

To analyze the impact of explainability regularization on the overall loss and the generated embeddings, we looked at the $R^2$ associated with the overall reconstruction error and the $R^2$ associated with the fitness of $M^o$, both as a function of regularization parameters. Table 1 shows $R^2$ results obtained for various values of the $\lambda_{aw}$. We can notice that the $R^2$ penalty for the reconstruction is maximized at $\lambda_{aw} = 0$. However, as $\lambda_{aw}$ increases, the $R^2$ remains mostly flat. In general, we noticed in our experiments that lower values of $\lambda_{aw}$ do not necessarily yield better $R^2$ despite reduction in entropy on the columns of $W$. This observation could be a manifestation of the Occam Razor principle that gives preferences for simpler and less complex models that focus on less variables. However, the penalty is much more severe on the goodness of fit of $M^o$ as we vary $\lambda_{aw}$. As $\lambda_{aw}$ increases, the complexity of the $M^o$ decreases as shown in Figure 5 but this reduction of complexity limits $M^o$'s ability to effectively estimate $M$. The explanations provided by the observer become trivial for the complexity of the task at hand.

## 4 CONCLUDING REMARKS

We have presented an MDL based approach to estimate the explainability of AI models while taking into account prior preferences. We have shown how these concepts can be used to regularize the

| $\lambda_{aw}$ | 0 | 1 | 2 | 3 | 4 | 5 |
|---|---|---|---|---|---|---|
| Avg $R^2$ between $\mathbf{X}$ and $\hat{\mathbf{X}}$ | 0.912 | 0.839 | 0.829 | 0.838 | 0.847 | 0.842 |
| Avg $R^2$ between $\mathbf{Y}$ and $\hat{\mathbf{Y}}$ | 0.997 | 0.861 | 0.672 | 0.461 | 0.434 | 0.386 |

Table 1: $R^2$ for different values of $\lambda_{aw}$

learning of temporal embeddings with regularization terms constraining the production of explainable models, according to metrics that we have defined. Experimental results on a real patient data set from MIMICIII has demonstrated the applicability of the proposed metrics for the generation of explainable models trading explanation complexity for overall accuracy. In the future, we plan to run further experiments on larger EHR data sets for other predictive tasks, with prior knowledge extracted from medical guidelines and ontologies. Furthermore, we plan to extend these concepts to reinforcement learning, in a policy-based setting where we constrain the learning of optimal policies with knowledge aware explainability metrics to produce explainable policies.

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
