# OpenReview forum: "ATTENTIVE EXPLAINABILITY FOR PATIENT TEMPORAL EMBEDDING"
_ICLR.cc/2019/Conference_

### Official Review · AnonReviewer3 · 2018-10-26
**A Definition for Interpretability based on MDL Principle**

**Rating:** 2
**Confidence:** 3

**Review:**

This paper proposes a definition for interpretability which is indeed the same as model simplicity using the MDL principle. It has several issues:

1) Interpretability is not the same as simplicity or number of model parameters. For example, an MLP is thought to be more interpretable than an RNN with the same number of parameters.

2) The definition of explainability in Eq. (5) is flawed. It should not have the second term L(M(X)|M^o, X) which is the goodness of M^o's fit. You should estimate M^o using that equation and then report L(M^o|M^p) as the complexity of the best estimate of the model (subject to e.g. linear class). Mixing accuracy of estimation of a model and its simplicity does not give you a valid explainability score.

3) In Section 2.4.2, the softmax operator will shrink the large negative coefficients to almost zero (reduces the degrees of freedom of a vector by 1). Thus, using softmax will result in loss of information. In the linear observer case, I am not sure why the authors cannot come up with a simple solution without any transformation.

4) Several references in the text are missing which hinders understanding of the paper.

---

> ### Author Response · Authors · 2018-11-27
> **Thank you for your valuable comments**
>
> Regarding issue 1) raised in your review, we would like to point out that we do not measure interpretability or explainability with a "number of model parameters". The use of the external observer treats the model as a blackbox. Our approach is in fact based on the premise that we do not want to inspect models to estimate explainability, let alone counting their number of parameters. Instead, we are training an external observer as described in the paper to estimate explainability.
>
> Regarding 2), we do not agree with the removal of the goodness of fit term. Removing it does not make much sense to us. Without this term, the observer would not be connected in any ways with the model M to be able to assess explainability of this model. Estimating L(M^o) or L(M^o|M^p) alone without any links to M does not make much sense. It is imperative to be able to make sure that the explanations provided fit the data produced by $M$. Consequently, we do not understand nor agree with the comment made in the review on the flaws found on this definition.
>
> Regarding 3), we have added a significant amount of details on how the softmax operator is used. The original submission was definitely omitting important details on the actual implementation of the scheme.
>
> Regarding 4), we have fixed these broken references and also addressed a few typos that we found in the text.
>
> Thank you for the review.

---

> > ### Comment · AnonReviewer3 · 2018-12-21
> > **Postmortem disagreement**
> >
> > Just letting you know that I had read your response but forgot to reply here.  I disagree with your response to 1 and 2, which is why I kept my score.

---

### Official Review · AnonReviewer1 · 2018-11-03
**Interesting approach, incomplete work.**

**Rating:** 3
**Confidence:** 4

**Review:**

Summary:
The authors propose a framework for training an external observer that tries explain the behavior of a prediction function using the minimal description principle. They extend this idea by considering how a human with domain knowledge might have different expectations for the observer. The authors test this framework on a multi-variate time series medical data (MIMIC-III) to show that, under their formulation, the external observer can learn interpretable embeddings.

Pros:
- Interesting approach: Trying to develop an external observer based on information theoretic perspective.
- Considering the domain knowledge of the human subject can potentially be an important element when we want to use interpretable models in practice.

Issues:
(1) In 2.4: So between M and M^O, which one is a member of M_reg?
(2) On a related note to issue (1): In 2.4.1, "Clearly, for each i, (M(X))_i | M^O(X) follows also a Gaussian distribution: First of all, I'm not sure if that expression is supposed to be  p(M(X)_i | M^O(X)) or if that was intended. But either way, I don't understand why that would follow a normal distribution. Can you clarify this along with issue (1)?
(3) In 2.4.2: The rationale behind using attention & compactness to estimate the complexity of M^O is weak. Can you elaborate this in the future version?
(4) What do each figure in Figure 4 represent?
(5) More of a philosophical question: the authors train M and M^O together, but it seems more appropriate to train an external observer separately. If we are going to propose a framework to train an agent that tries to explain a black box function, then training the black-box function together with the observer can be seen as cheating. It can potentially make the job of the observer easier by training the black box function to be easily explainable. It would have been okay if this was discussed in the paper, but I can't find such discussion.
(6) The experiments can be made much stronger by applying this approach to a specific prediction task such as mortality prediction. The current auto-encoding task doesn't seem very interesting to apply interpretation.
(7) Most importantly: I like the idea very much, but the paper clearly needs more work. There are broken citations and typos everywhere. I strongly suggest polishing this paper as it could be an important work in the model interpretability field.

---

> ### Author Response · Authors · 2018-11-27
> **Thank you for the detailed review. Your feedback is greatly appreciated.**
>
> We have attempted to address your comments in the following way.
> Regarding issue (1) above, we have added text in 2.4 to address your valid concern. It turns out that for the MDL, we are really interested in M^o. However, the regression restriction is really dictated by the problem that we are trying to solve. Both M and M^o belong to M_reg.
>
> Regarding (2), there was a typo that we fixed as you pointed (missing "p"). Regarding the gaussian assumption, this is pretty standard when trying to model the distribution of a regression error. It is a standard assumption for which we added a pointer in the paper to a text book on MDL.
>
> Regarding 3, compactness simply relates to description lengths. A more compact model will require less bits to be described. As stated in the paper, the MDL does not really mandate how the model complexity should be computed. Attention models have been suggested for explainability in the literature. We are expanding on these approaches. More complex models can certainly be looked and we plan to do this in the future.
>
> Regarding 5, the gist of this work is to develop metrics of explainability to: (i) be able to estimate or measure how explainable deep learning models are and (ii) to be able to force the learning of deep models towards models that are easy to explain. For (i), training an external observer separately makes sense. We could have reported results experimental results on this since the code that we have does it implicitly to estimate these metrics during the learning. Our focus in the paper is mostly on (ii), a harder problem in our view. We do not view this joint learning as "cheating" (it could just be a poor choice of words). We are only trying to enforce explainability constraints on the training of architectures like RNNs that are notoriously hard to explain. The joint learning with the observer allows us to do just that. There is a delicate balance between the expressiveness of the model M and explainability enforced by the observer M^o. If the black box is too easy to explain, it will not perform well on its task. Similarly, if the backbox is very complex, it will not be easy to explain. The joint learning helps us trade between these extremes, as shown by the experiments where we vary the hyper-parameters controlling this trade-off. There is a discussion on this trade-off in the experimental section.
>
> Regarding 6, we agree that this approach can be applied on other deep learning tasks. We have decided to use this embedding problem in this paper simply because applications of deep learning techniques to these problems in healthcare is hindered by explainability requirements. We definitely plan on applying the framework on other problems in the future.
>
> Regarding 7, we do agree and have taken multiple passes at it to address these issues.
>
> Finally, we removed Figure 4. It was a bit distractive. It was meant to show how temporal attention coefficients were distributed at the decoder side.
>
> Many thanks for the very constructive comments.

---

### Official Review · AnonReviewer2 · 2018-11-04
**Interesting problem and hypothesis, inconclusive analyses and experiments**

**Rating:** 4
**Confidence:** 4

**Review:**

This paper is motivated in an interesting application, namely "explainable representations" of patient physiology, phrased as a more general problem of patient condition monitoring. Explainability is formulated as a communication problem in line with classical expert systems (http://people.dbmi.columbia.edu/~ehs7001/Buchanan-Shortliffe-1984/MYCIN%20Book.htm).
Information theoretical concepts are applied, and performance is quantified within the minimum description length (MDL) concept.

Quality & clarity
While the patient dynamics representation problem and the communication theoretical framing is interesting , the analyses and experiments are not state of the art.
While the writing overall  is clear and the motivation well-written, there are many issues with the modeling and experimental work.
The choice of MDL over more  probabilistic approaches  (as e.g. Hsu et al 2017 for sequences) could have been better motivated. The attention mechanism could have been better explained (attention of whom and to what?) and also the prior (\beta). How is the prior established - e.g. in the MIMIC case study
The experimental work is carried out within a open source data set - not allowing the possibility of testing explanations against experts/users.

Originality
The main originality is in the problem formulation.

Significance
The importance of this work is limited as the case is not clearly defined. How are the representations to be used and what type of users is it intended to serve (expert/patients etc)

Pros and cons
+ interesting problem

-modeling could be better motivated
-experimental platform is limited for interpretability studies

==
Hsu, W.N., Zhang, Y. and Glass, J., 2017. Unsupervised learning of disentangled and interpretable representations from sequential data. In Advances in neural information processing systems (pp. 1878-1889).

---

> ### Author Response · Authors · 2018-11-27
> **Thank you for the comments, much appreciated**
>
> We have addressed your feedback by taking a full pass at the entire manuscript. We specially focused on rewriting many aspects of the methodology. For instance, we have added text explaining the use of attention mechanism and detailing how they are computed as part of the observer model.
> Many thanks.

---

### Meta-Review · Area_Chair1 · 2018-12-14

**Confidence:** 4
**Recommendation:** Reject

**Metareview:**

The paper proposes an approach to define an "interpretable representation",
in particular for the case of patient condition monitoring. Reviewers point
to several concerns, including even the definition of explainability and
limited significance. The authors tried to address the concerns but reviewers
think the paper is not ready for acceptance. I concur with them in rejecting it.